# ONE SIZE DOESN'T FIT ALL: ADAPTIVE LABEL SMOOTHING

## ABSTRACT

This paper concerns the use of objectness measures to improve the calibration performance of Convolutional Neural Networks (CNNs). CNNs have proven to be very good classifiers and generally localize objects well; however, the loss functions typically used to train classification CNNs do not penalize inability to localize an object, nor do they take into account an object's relative size in the given image. During training on ImageNet-1K almost all approaches use random crops on the images and this transformation sometimes provides the CNN with background only samples. This causes the classifiers to depend on context. Context dependence is harmful for safety-critical applications. We present a novel approach to classification that combines the ideas of objectness and label smoothing during training. Unlike previous methods, we compute a smoothing factor that is *adaptive* based on relative object size within an image. This causes our approach to produce confidences that are grounded in the size of the object being classified instead of relying on context to make the correct predictions. We present extensive results using ImageNet to demonstrate that CNNs trained using adaptive label smoothing are much less likely to be overconfident in their predictions. We show qualitative results using class activation maps and quantitative results using classification and transfer learning tasks. Our approach is able to produce an order of magnitude reduction in confidence when predicting on context only images when compared to baselines. Using transfer learning, we gain $0.021AP$ on MS COCO compared to the hard label approach.

## 1 INTRODUCTION

Convolutional neural networks (CNNs) have been used for addressing many computer vision problems for over 2 decades (LeCun, 1998); in particular, showing promising results on object detection and localization tasks since 2013 (Krizhevsky et al., 2012; Russakovsky et al., 2015; Girshick et al., 2018). Unfortunately, modern CNNs are overconfident in their predictions (Lakshminarayanan et al., 2017; Hein et al., 2019) and they suffer from reliability issues due to miscalibration (Guo et al., 2017a). Problems related to overconfidence, generalization, bias and reliability represent a severe limitation of current CNNs for real-world applications. We address the problems of overconfidence and contextual bias in this work.

Recently, (Szegedy et al., 2016) introduced label smoothing, providing soft labels that are a weighted average of the hard targets uniformly distributed over classes during training, to improve learning speed and generalization performance. In the case of classification CNNs, ground-truth labels are typically provided as a one-hot (hard labels) representation of class probabilities. These labels consist of 0s and 1s, with a single 1 indicating the pertinent class in a given label vector. Label smoothing minimizes weight magnification (Mukhoti et al., 2020; Müller et al., 2019) and shows improvement in learning speed and generalization; in contrast, hard targets tend to increase the values of the logits and produce overconfident predictions (Szegedy et al., 2016; Müller et al., 2019). Label smoothing and the traditional hard labels force CNNs to produce high confidence predictions even when pertinent objects are absent during training. To obtain more reliable confidence measures, we use the objectness measure to derive a smoothing factor for every sample undergoing a unique scale and crop transformation in an adaptive manner. Safely deploying deep learning based models has also become a more immediate challenge (Amodei et al., 2016). As a community, we need to obtain high accuracies, but also provide reliable uncertainty measures of CNNs. We can improve the precision

of CNNs by providing reliable confidence measures, avoiding acting with certainty when uncertain predictions are produced, as in the case of safety-critical systems.

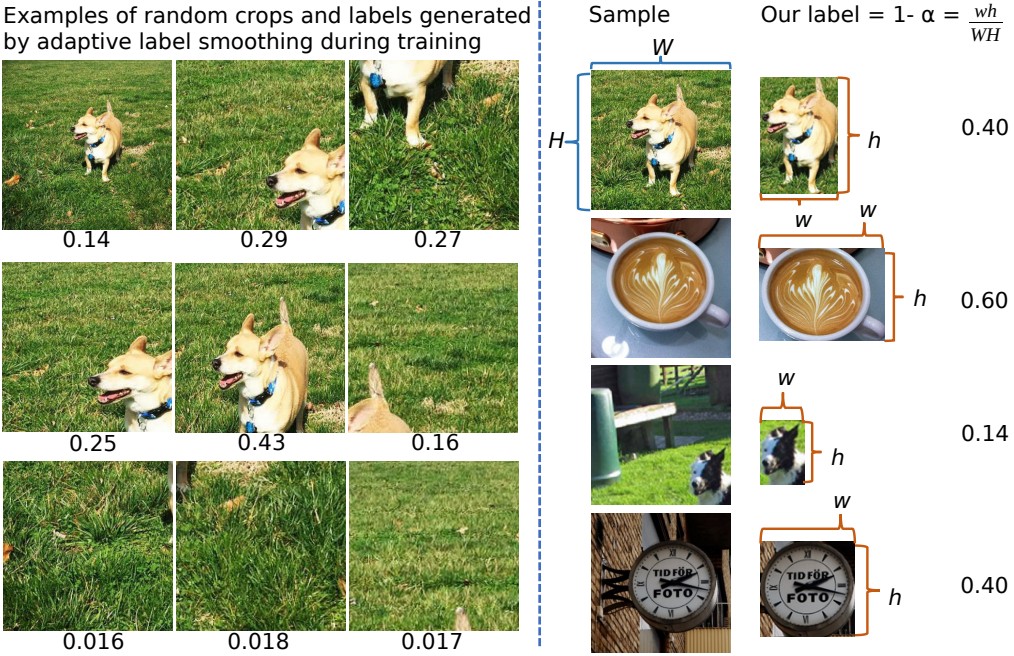

Figure 1: Random crops of images are often used when training classification CNNs to help mitigate size, position and scale bias (as shown in the left half of the figure along with the objectness values listed below them). Unfortunately, some of these crops miss the object as the process does not use any object location information. Traditional hard label and smooth label approaches do not account for the proportion of the object being classified and use a fixed label of 1 or 0.9 in the case of label smoothing. Our approach (right half) smooths the hard labels by accounting for the objectness measure to compute an *adaptive* smoothing factor. The objectness is computed using bounding box information as shown above. Our approach helps generate accurate labels during training and penalizes low-entropy (high-confidence) predictions for context-only images.

Object detection (Girshick et al., 2018) requires bounding box information during training. Recently, (Dvornik et al., 2018) proposed using novel synthetic images to improve object detection performance by augmenting training data using object location information; however, classification CNNs have not exploited object size information to regularize CNNs on large datasets like ImageNet (Russakovsky et al., 2015), to our knowledge. Objectness, quantifying the likelihood an image window contains an object belonging to any class, was first introduced by (Alexe et al., 2012), and the role of objectness has been studied extensively since then. Object detectors specialize in a few classes, but objectness is class agnostic by definition. We limit the definition of objectness to the 1000 ImageNet-1K classes, meaning any object outside these defined classes will have an objectness score of 0. When training a classifier, the cross-entropy loss is employed but it does not penalize incorrect spatial attention, often making CNNs overfit to context or texture rather than the pertinent object (Geirhos et al., 2019), as shown in the left half of figure 1. The bottom row displays samples with negligible amounts of 'Dog' pixels, where traditional methods would label them as 'Dog', causing CNNs to output incorrect predictions with high confidence when presented with images of backgrounds or just context. Adaptive label smoothing (our approach) involves using gross object size to smooth the hard labels of a classifier, as displayed to the right in figure 1. Our approach adapts label smoothing by deriving the smoothing factor using the objectness measure. When compared to approaches based on hard labels, sample mixing, and label smoothing, our approach improves object detection and calibration performance. Traditional approaches (Yun et al., 2019; Takahashi et al., 2018; Krizhevsky et al., 2012; Russakovsky et al., 2015) use random resize and random crop augmentation, and sometimes lose the pertinent object in the training sample, allowing the classifier to make the correct predictions by overfitting to the context surrounding the pictures. We believe

that our approach addresses significant problems that are associated with current training techniques. In particular, random cropping of images is a common augmentation technique during training of classifiers, but occasionally the crop misses the object entirely. In such a case, the equivalent of a one-hot label is typically provided, with the result that the system is steered toward increased dependence on background (context) portions of the image. We argue that one-hot representations are too limiting, and our adaptive approach to label smoothing makes it possible for the classifier to avoid overconfidence in many cases. Specifically, our contributions are listed below:

1. Our regularization technique, called *adaptive label smoothing* adjusts labels during training based on an object's relative size for every sample, directly affecting the confidence measure produced by the classifier. This implicit regularization guides the classifiers, avoiding high confidence predictions when the object pixels are lower in proportion.

2. For safety-critical applications, our approach allows the classifier to produce low confidence predictions when images with context and no pertinent object are presented. Predictions from our approach are more explainable and they can easily be thresholded to reject false positives. High confidence approaches are hard to threshold as predictions have high confidence, even when they are wrong. Our predictions are more explainable as the confidence is grounded in object size and not context. We assume that every class is equiprobable when inputs without pertinent objects are supplied during training. While context helps increase computed accuracy for a given dataset, such reliance is not viable for real-world applications.

3. We show that the representation learned with adaptive label smoothing also leads to better transfer learning performance on MS COCO (Lin et al., 2014).

We have trained classifiers and evaluated them on three popular datasets, with results showing our approach produces an average confidence that is an order of magnitude lower when compared to baselines for context-only images. Confidence values generated by CNNs help us understand the output predictions, but unreliable confidence measures hurt the applicability of CNNs for safety-critical applications.

## 2 RELATED WORK

Bias exhibited by machine learning models can be attributed to many underlying statistics present in datasets and model architectures (Battaglia et al., 2018; Zhang et al., 2017b) including context, object texture (Geirhos et al., 2019), size, shape, and color in the case of images. Various approaches to mitigate bias have been proposed (Anne Hendricks et al., 2018; Choi et al., 2019; Geirhos et al., 2019) in recent years. Our approach produces high entropy predictions when context-only images are provided as input during inference, as we aim to learn the size of the relevant object within the image and classify it, instead of relying on contextual bias to produce a prediction.

Traditionally, any label preserving transformation on an input image is employed to help regularize a CNN. The authors of AlexNet (Krizhevsky et al., 2012) employed random cropping and horizontal flipping methods, designed to prevent overfitting and improve the generalization of viewpoints when they surpassed the performance of conventional machine learning approaches in 2012. The random noise class of data augmentation methods (DeVries & Taylor, 2017; Zhong et al., 2017) mask random regions of an input image with zeros, which may accidentally erase the pertinent object in a given image forcing the CNN to rely on context to make a prediction, contributing to label noise. The authors of DropBlock (Ghiasi et al., 2018) have used dropout in the feature space to obtain better generalization. Authors of AutoAugment (Cubuk et al., 2019) used reinforcement learning dynamically during training to learn the best combination of existing data augmentation methods. The latest work in the area of data augmentation uses samples from different classes and changes expected outputs to predict a probability distribution based on the number and intensity of pixels represented by each class. The authors of Mixup (Zhang et al., 2017a; Tokozume et al., 2018) use alpha blending (weighted sum of pixels from two different classes) applied to corresponding labels. The authors of CutMix and RICAP (Yun et al., 2019; Takahashi et al., 2018) use soft labels by cropping different regions and classes of images and 'mixing' the labels proportionally to corresponding regions in the final augmented sample. Neither of these approaches relies on object size when 'mixing' regions in images and computing the label. Conversely, our approach regularizes classification CNNs by using objectness information and applying a smoothing factor based on an object's proportion in a given image, producing a soft label without mixing the samples.

Calibration and uncertainty estimation of predictors has been an ongoing interest to the machine learning community (Murphy, 1973; DeGroot & Fienberg, 1983; Platt, 1999; Lin et al., 2007; Zadrozny & Elkan, 2002) as predictions need to be equally accurate and confident. Bayesian binning into quantiles (BBQ) (Naeini et al., 2015) was proposed for binary classification and beta calibration, and (Kull et al., 2017) employed logistic calibration for binary classifiers. In the context of CNNs, (Guo et al., 2017b) proposed a temperature scaling approach to improve calibration performance of pre-trained models. Calibration has been explored in multiple directions; popular approaches include transforming outputs of pre-trained models using approximate bayesian inference (Maddox et al., 2019), or using a special loss function to help regularize the model (Pereyra et al., 2017; Kumar et al., 2018) during training. Our approach is loosely related to the latter class of methods and relates to label smoothing proposed first by (Szegedy et al., 2016), with applicability for many tasks explored by (Pereyra et al., 2017); (Xie et al., 2016) applies dropout like noise to the labels. Recently, (Müller et al., 2019) explored the benefits of label smoothing; aside from having a regularizing effect, label smoothing helps reduce the intra-class distance between samples (Müller et al., 2019). Another approach to calibrate CNNs was proposed by (Mukhoti et al., 2020). By applying a focal loss function and temperature scaling, the authors were able to obtain state-of-the-art calibration performance. Label smoothing also improves calibration performance of CNNs (Mukhoti et al., 2020).

In contrast to previously discussed methods, our approach involves using hard labels multiplied by the objectness measure and obtaining a uniform distribution over all other classes when input images are devoid of pertinent objects. We do not change our loss function as opposed to (Mukhoti et al., 2020) or add additional layers to our model. Our approach can be described as a variant of label smoothing, employing an *adaptive* label smoothing approach that is unique to every training sample as it accounts for object size. To our knowledge, we are the first to apply objectness based *adaptive* label smoothing to train image classification CNNs. The objectness is computed using bounding box information during training. CNNs trained using hard labels produce 'peaky' probability distributions without considering the spatial size of the pertaining object. Our approach produces outputs that are softer and the peaks correspond to the spatial footprint of the object being classified as illustrated in the appendix A.1.

## 3 METHOD

We provide a mathematical discussion of the cross entropy loss computed using different approaches in this section. Consider $D = \langle (\mathbf{x}_i, y_i) \rangle_{i=1}^N$ to be a dataset consisting of $N$ independent and identically distributed real-world images belonging to $K$ different classes. Let $\mathcal{X}$ represent the set of images, and let $\mathcal{Y}$ denote the set of ground-truth class labels. Sample $i$ consists of the image $\mathbf{x}_i \in \mathcal{X}$ along with its corresponding label $y_i \in \mathcal{Y} = \{1, 2, ..., K\}$. Let $f_\theta$ represent the CNN classifier $f$ with model parameters denoted by $\theta$. The predicted class is $\hat{y}_i = \text{argmax}_{y \in \mathcal{Y}} \hat{p}_{i,y}$, where $\hat{p}_{i,y} = f_\theta(y|\mathbf{x}_i)$ is the computed probability that the image $\mathbf{x}_i$ belongs to the class $y$. The confidence or class probability can be computed using $\hat{p}_i = \max_{y \in \mathcal{Y}} \hat{p}_{i,y}$, following the notation adopted by (Mukhoti et al., 2020).

We denote the output probability distribution over $K$ classes after applying the softmax function as:

$$\hat{p}_{i,y} = \frac{\exp(p_k)}{\sum_{j=1}^K \exp(p_j)} \tag{1}$$

where, $p_k$ represents the logit for class $k$. Let $\pi(k|x_i)$ be the one-hot label vector ($K$-element long) corresponding to input $x_i$ and $k \in \{1, 2, ..., K\}$. The cross-entropy loss $\mathcal{L}$ used to train the CNN is computed by:

$$\mathcal{L}(x_i) = -\sum_{j=1}^K \pi(k|x_i) \log(\hat{p}_{i,y}) \tag{2}$$

In the case of one-hot labels, $\pi(y|x_i) = 1$ for the pertinent class $y$ and $\pi(k|x_i) = 0$ for all other classes $k \neq y$. The cross entropy loss can now be reduced to a single term as opposed to a summation:

$$\mathcal{L}(x_i) = -\log(\hat{p}_{i,y}) \tag{3}$$

There are three problems associated with the loss described above:

1. The CNN is encouraged to produce a very large peak for the pertinent class $y$ and the CNN is not penalized for producing peaks for incorrect classes, $k \neq y$.

2. The supplied label and input may not always be correct when random cropping is used during training. More precisely, predicting correctly or incorrectly with high confidence based on just context shows that random cropping can lead to overreliance on context (predicting the presence of a dog based on an image of a dog park without any dogs, for example).

3. The CNNs trained with one-hot labels produce extremely high confidence values ($\hat{p}_i$) without paying attention to the presence of an object or its proportion.

Following (Szegedy et al., 2016), the hard label $\pi(k|x_i)$ can be converted to soft label $\tilde{\pi}(k|x_i)$ using $\tilde{\pi}(k|x_i) = \pi(k|x_i)(1-\alpha) + (1-\pi(k|x_i))\alpha/(K-1)$, where $\alpha \in [0, 1]$ is a fixed hyperparameter. This is the standard procedure known as label smoothing or uniform label smoothing. The cross-entropy loss $\mathcal{L}_{ls}$ for uniform label smoothing can be written as:

$$\mathcal{L}_{ls}(x_i) = -(1-\alpha)\log(\hat{p}_{i,y}) - \alpha(\sum_{j \neq y}^{K}(1 - \pi(k|x_i))\log(\hat{p}_{i,y}))/(K-1) \tag{4}$$

The novelty of our approach is to make $\alpha$ *adaptive*, calculating the value based on the relative size of an object within a given training image. Using the bounding box annotations available for the images in the dataset, we generate object masks. We apply the same augmentation transform (scale, crop) to the masks and compute the objectness score on the fly for every training image. Let the image width and height be denoted by $(W, H)$ and the object width and height be denoted by $(w, h)$. The ratio $\alpha$ is computed as $\alpha = 1 - \frac{wh}{WH}$. The soft label $\tilde{\pi}(k|x_i)$ is computed as before:

$$\tilde{\pi}(k|x_i) = \pi(k|x_i)(1-\alpha) + (1-\pi(k|x_i))\alpha/(K-1) \tag{5}$$

We also explore a weighted combination of *adaptive* label smoothing and hard labels. To do this, we introduce parameter $\beta \in [0, 1]$ to determine the degree of *adaptive* label smoothing being applied. The setting $\beta = 0$ corresponds to the case of classic hard labels. The soft label in this case is computed as $\tilde{\pi}(k|x_i) = (\pi(k|x_i)(1-\alpha) + (1-\pi(k|x_i))\alpha/(K-1))\beta + (1-\beta)(\tilde{\pi}(k|x_i))$. The cross-entropy loss $\mathcal{L}_{als}$ with adaptive label smoothing can be written as:

$$\mathcal{L}_{als}(x_i) = -\beta((1-\alpha)\log(\hat{p}_{i,y}) - \alpha(\sum_{j \neq y}^{K}(1 - \pi(k|x_i))\log(\hat{p}_{i,y}))/(K-1)) - (1-\beta)\log(\hat{p}_{i,y}) \tag{6}$$

We penalize the CNN for producing high confidence predictions when the objectness score is low using an adaptive $\alpha$. We introduce $\beta$ as an ablation parameter to adjust the amount of context dependence allowed. When $\beta$ is set to 0, we end up with one-hot labels and when $\beta$ is set to 1, the CNN is trained using adaptive label smoothing. Setting a value of $\beta$ above 0 (under 1) reduces the context dependence. When $\beta$ is set to 0.75, the CNN is trained with a label of at least 0.25 for the pertinent class regardless of whether an object is present or not. The rest of the label is computed using adaptive label smoothing and weighted by $\beta$. As adaptive label smoothing accounts for object size, the label for the pertinent class will increase based on the objectness score for the sample. When $K$ is small $\beta$ can be adjusted to avoid computing incorrect labels for objects with low objectness score.

## 4    EXPERIMENTS

In this section, we provide a description of the datasets used in our experiments, introduce some of the commonly used metrics for calibration of CNNs and describe our implementation details. We then discuss the merits of our approach and answer important questions related to applicability to transfer learning in an object detection setting, and we discuss the effect of using different types of labels during training in an ablative manner. We use ResNet-50 (He et al., 2016) for most of our experiments and ResNet-101 (He et al., 2016) for the rest. For additional information on experimental setup please refer to the appendix A.3.

## 4.1 DATASETS

We have used different training datasets that are based on ImageNet-1K dataset (Russakovsky et al., 2015). ImageNet-1K consists of 1.28M training images and 50K validation images spanning 1K categories. As only 38% of ImageNet training images have bounding-box annotations, we distinguish these experiments from those trained on the full dataset. We use standard data-augmentation strategies for all methods and train all our models for 300 epochs starting with a learning rate of 0.1 and decayed by 0.1 at epochs 75, 150, and 225 using a batch size of 256. As shown in tables A.7, we have different training datasets that are based on ImageNet-1K dataset (Russakovsky et al., 2015). For additional dataset information please refer to the appendix A.2.

Our method needs object proportions to compute the objectness score, we use a subset of the standard ImageNet dataset that has bounding boxes (0.474M). To generate the 'mask' version, we make sure that only one object is present in a given image and 'mask' all other objects replacing them with pixel means. We use this version of the dataset derived from the 0.474M subset and identify the approach with '(mask)' next to the method in tables 2 and 3. We end up with about 54K more images as some ImageNet images have multiple annotated objects and our training dataset has 0.528M images as a result. Lastly, we generate another dataset that is devoid of any object altogether. We sample about 15% of the time from this dataset during training of one (identified with 'Context') of our approaches, and the label generated for these methods is a vector of uniform probability distribution across 1000 classes. The idea is that when no objects are present in a sample, a CNN should produce a high-entropy prediction.For validation, we use the validation set of (Russakovsky et al., 2015) (V1) and the newly released ImageNetV2 set (Recht et al., 2019). Specifically, we use the more challenging 'MatchedFrequency' set of images. The different validation sets are identified in the 'Val.' column of table 2.

To measure the transfer-learning ability of the representations learned by our classifiers, we used the challenging MS COCO (Lin et al., 2014) dataset to obtain the results described in table 4. The dataset consists of about 230K training images and we use the 'minival' validation set of 5K images with bounding box annotations.

## 4.2 CLASSIFICATION, CONTEXT AND CALIBRATION

This section identifies various calibration metrics used by the community and discusses our results obtained on the popular (Russakovsky et al., 2015; Recht et al., 2019) datasets. We use the implementation of (Wenger et al., 2020) on all of our classifiers to generate the results in table 2.For extended results, refer to appendix A.7. To evaluate the performance of *adaptive* label smoothing we use five metrics that are very common: *accuracy* (ACC), *expected calibration error* (ECE) (Naeini et al., 2015), *maximum calibration error* (MCE) (Naeini et al., 2015), *overconfidence* (Mund et al., 2015), and *underconfidence* (Mund et al., 2015). We computed ECE using 100 bins and 15 bins. The authors of (Wenger et al., 2020; Kumar et al., 2019) discuss the advantages of using 100 bins in greater detail.

A classifier is said to be calibrated if its confidence matches the probability of the prediction being correct, $\mathbb{E}\left[1_{\hat{y}_i=y_i} \mid \hat{p}_i\right] = \hat{p}_i$. ECE is defined as the expected absolute difference between a classifier's confidence and its accuracy using a finite number of bins (Naeini et al., 2015; Wenger et al., 2020). ECE is computed as, $\text{ECE} = \mathbb{E}\left[|\hat{p}_i - \mathbb{E}\left[1_{\hat{y}_i=y_i} \mid \hat{p}_i\right]|\right]$. MCE is defined as the maximum absolute difference between a classifier's confidence and its accuracy of each bin (Naeini et al., 2015; Wenger et al., 2020), MCE is computed as, $\text{MCE} = \max_{\hat{p}_{i,y}\in[0,1]}|\hat{p}_i - \mathbb{E}\left[1_{\hat{y}_i=y_i} \mid \hat{p}_i = \hat{p}_{i,y}\right]|$.

Overconfidence is the average confidence of a classifier's false predictions, mathematically computed as, $o(f) = \mathbb{E}\left[\hat{p}_i \mid \hat{y}_i \neq y_i\right]$. Underconfidence is the average uncertainty on its correct predictions (Wenger et al., 2020; Mund et al., 2015), mathematically computed as, $u(f) = \mathbb{E}\left[1 - \hat{p}_i \mid \hat{y}_i = y_i\right]$. Overconfidence and underconfidence of a classifier are not reflective of its accuracy (Wenger et al., 2020).

Our approach uses labels that are more accurate than other baselines when random cropping and scaling of images are applied during training. To our knowledge, almost all classifiers trained on ImageNet use random crop and scaling based augmentation to regularize. The random crop transformation allows the CNNs to predict by relying on context rather than the pertinent object, our approach uses bounding box labels to produce labels in an adaptive way during training. To quantify context dependence, we used bounding box annotations on the 50K validation images, removed all

Table 1: Confidence and accuracy metrics on the validation set of ImageNet with all the objects removed using bounding box annotation provided by Choe et al. (2020). Our approach has the best performance under total uncertainty. 'ACC', 'A.conf', 'O.conf' and 'U.conf' refer to accuracy, average confidence, mean overconfidence, and mean underconfidence scores. High underconfidence and low overconfidence point to minimal reliance on context when no pertinent objects are in the given image. The last row of figure 6 in appendix provides a qualitative example.

| Method | ACC | O.conf | U.conf | A.conf |
|---|---|---|---|---|
| Hard Label | 0.0633 | 0.2734 | 0.3362 | 0.2982 |
| Label Smoothing (Szegedy et al., 2016) | 0.0618 | 0.1851 | 0.4816 | 0.2057 |
| CutMix (Yun et al., 2019) | 0.0921 | 0.1679 | 0.4696 | 0.2013 |
| A. L. S. (Ours) | **0.0473** | **0.0121** | **0.8409** | **0.0191** |

objects and replaced the pixels with the mean image pixel values using bounding box annotation provided by (Choe et al., 2020). Hard label trained CNN had an accuracy of 6.3% with an average confidence of 0.29, label smoothing based CNN predicted with an accuracy of 6.1% and an average confidence of 0.2, CutMix had an accuracy of 9.2% with an average confidence of 0.2. These baseline methods produced high confidence predictions on images with no objects present using just context information. Our approach had an accuracy of 4.7% and an average confidence of 0.02. We have an order of magnitude improvement in performance over recent baselines as our approach helps CNNs produce confidence based on the relative size of the pertinent object. Predictions using our method are more explainable as we ground our labels and confidences in the object size, as opposed to making correct predictions using contextual information only as shown in table 1. The results in table 2 indicate our approaches based on *adaptive* label smoothing using the abbreviation 'A. L. S.' In general, these results have a low overconfidence score. This is highly desirable for safety-critical applications, as when our approach is wrong, it is wrong with the least amount of confidence. When no pertinent objects are present, our approach is the *least* confident compared to other baselines this makes our approach more suitable for safety-critical applications.

The mean objectness of images in the validation set of ImageNet is 0.49. The mean objectness deviation, computed as the mean of the absolute difference between maximum confidence and objectness over the ImageNet validation samples, for our approach is 0.24 as opposed to 0.42 for the hard label case. Using these metrics, we show that our confidences are more explainable as they closely match the objectness statistics when compared to the hard label approach. Our approach is underconfident as we are not trying to produce the maximum possible confidence of 1 when we are correct. Our confidence is grounded in the objectness score instead, our peaks are proportional to the size of the object. These results demonstrate that adaptive label smoothing based CNNs seldom produce high confidence scores when they make incorrect predictions. In fact, our models are underconfident as they pay attention to the spatial footprint of the pertinent object instead of producing a large peaks most of the time. It is important to note that our methods outperform all baselines for the overconfidence metric. ECE and MCE measure the difference between a classifier's accuracy and its prediction, our approach has higher values as our predictions are not the same as the accuracy of the classifier. As we intend to produce peaks proportional to objectness values instead of the classifier's accuracy. As shown in figure 2, we are over the diagonal, *we are more accurate than we are confident compared to baselines*.

### 4.3 Transfer Learning for Object Detection

We use the MS COCO (Lin et al., 2014) dataset to benchmark our transfer learning performance. We adopt the architecture of Faster RCNN (Ren et al., 2015) adapted to use the ResNet-50 backbone. Specifically, we train all of our classifiers using the implementation of *https://github.com/jwyang/faster-rcnn.pytorch*. We train all ImageNet pre-trained models with a batch size of 16 and initial learning rate of 0.01 decayed after every 4 epochs for a total of 10 epochs. We employ the standard metrics for average precision (AP) and average recall (Lin et al., 2014) at different intersection over union (IoU) levels. As shown in 4, our approach outperforms hard label and label smoothing based approaches on this downstream task. Specifically, our approach performs almost as well as CutMix (Yun et al., 2019) using AP measures. For information on qualitative

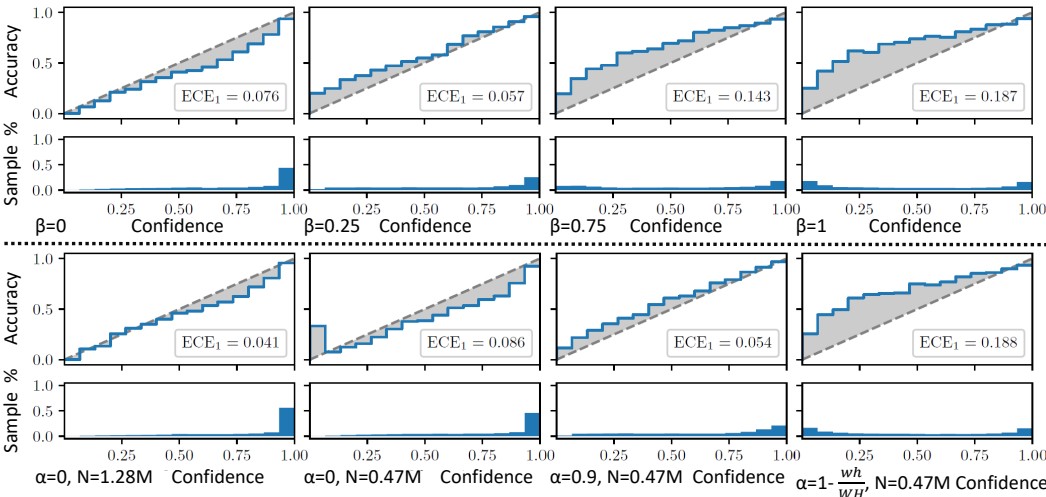

Figure 2: Reliability diagrams help understand the calibration performance (DeGroot & Fienberg, 1983; Niculescu-Mizil & Caruana, 2005) of classifiers. We compute $ECE_1$ using the implementation of (Wenger et al., 2020) on the validation set of ImageNet. The deviation from the dashed line (shown in gray), weighted by the histogram of confidence values, is equal to Expected Calibration Error (Wenger et al., 2020). The top half of the figure shows classifiers trained using the same dataset ($N$=0.528M), but with different values of $\beta$. The leftmost reliability diagram is the classic hard label setting and the rightmost reliability diagram is the *adaptive* label setting. The bottom half of the figure compares classifiers trained on the complete ImageNet (leftmost) with 3 classifiers trained on the subset of ImageNet with bounding box labels using different values of the $\alpha$ hyperparameter.

localization performance without any fine-tuning using class activation maps, please refer to appendix A.6.

Table 2: Classification and calibration results with ImageNet using ResNet-50. For a detailed explanation of the metrics please refer to 4.2.'O.conf' and 'U.conf' refer to overconfidence and underconfidence scores. Our approach produces low overconfidence values. We use the single object version (mask) of ImageNet with 0.528M samples to train all the models below.

| Method | Val. Set | ACC | ECE 100 | ECE 15 | MCE | O.conf | U.conf |
|---|---|---|---|---|---|---|---|
| Hard Label (mask) (Beta=0) | V1 | 0.680 | 0.088 | 0.076 | 0.259 | 0.549 | 0.132 |
| Hard Label (mask) (Beta=0) | V2 | 0.559 | 0.155 | 0.127 | 0.686 | 0.507 | 0.163 |
| CutMix (mask) | V1 | 0.698 | 0.032 | 0.020 | 0.249 | 0.477 | 0.197 |
| CutMix (mask) | V2 | 0.576 | 0.110 | 0.067 | 0.614 | 0.449 | 0.228 |
| Label Smoothing (mask) | V1 | 0.687 | 0.051 | 0.046 | 0.430 | 0.407 | 0.244 |
| Label Smoothing (mask) | V2 | 0.563 | 0.108 | 0.048 | 0.524 | 0.374 | 0.281 |
| A. L. S. (mask) (Beta=1) | V1 | 0.648 | 0.186 | 0.182 | 0.463 | 0.246 | 0.396 |
| A. L. S. (mask) (Beta=1) | V2 | 0.528 | 0.185 | 0.160 | 0.687 | 0.209 | 0.441 |
| A. L. S. (mask) (Beta=0.75) | V1 | 0.681 | 0.146 | 0.142 | 0.377 | 0.319 | 0.337 |
| A. L. S. (mask) (Beta=0.75) | V2 | 0.556 | 0.146 | 0.113 | 0.572 | 0.274 | 0.375 |
| A. L. S. (mask) (Beta=0.25) | V1 | 0.684 | 0.059 | 0.052 | 0.244 | 0.402 | 0.244 |
| A. L. S. (mask) (Beta=0.25) | V2 | 0.561 | 0.109 | 0.059 | 0.627 | 0.369 | 0.285 |
| A. L. S.+Context (mask) | V1 | 0.637 | 0.174 | 0.169 | 0.431 | 0.251 | 0.390 |
| A. L. S.+Context (mask) | V2 | 0.515 | 0.177 | 0.147 | 0.682 | 0.221 | 0.437 |

## 4.4 ABLATION STUDIES

We compare our approach with standard baselines and provide results in an ablative manner to understand the benefits and limitations of applying *adaptive* label smoothing to classification and

Table 3: Classification and calibration results with ImageNet using ResNet-101. For a detailed explanation of the metrics please refer to 4.2.'O.conf' and 'U.conf' refer to overconfidence and underconfidence scores.

| Method | Val. Set | ACC | ECE 100 | ECE 15 | MCE | O.conf | U.conf |
|---|---|---|---|---|---|---|---|
| Hard Label (mask) | V1 | 0.698 | 0.088 | 0.076 | 0.264 | 0.566 | 0.119 |
| Hard Label (mask) | V2 | 0.584 | 0.157 | 0.129 | 0.708 | 0.538 | 0.151 |
| A. L. S. (mask) | V1 | 0.669 | 0.145 | 0.138 | 0.408 | 0.319 | 0.315 |
| A. L. S. (mask) | V2 | 0.540 | 0.157 | 0.122 | 0.657 | 0.281 | 0.354 |

Table 4: Fine-tuning on MS COCO using FRCNN for object detection using ResNet-50 backbone. For a detailed explanation of the results please refer to 4.3. AP refers to average precision and AR refers to average recall at the specified Intersection over union (IoU) level. Our AP is only 0.001 lower than CutMix.

| Method | AP (0.5:0.95) | AP (0.5) | AP (0.75) | AR (0.5:0.95) |
|---|---|---|---|---|
| Hard Label (mask) | 0.290 | 0.482 | 0.307 | 0.415 |
| CutMix (mask) | 0.312 | 0.509 | 0.329 | 0.428 |
| Label Smoothing (mask) | 0.304 | 0.500 | 0.324 | 0.424 |
| A. L. S. (mask) | 0.311 | 0.501 | 0.333 | 0.428 |
| A. L. S. (mask) (Beta=0.75) | 0.309 | 0.498 | 0.331 | 0.427 |
| A. L. S. (mask) (Beta=0.25) | 0.298 | 0.492 | 0.315 | 0.419 |
| A. L. S. + Context (mask) | 0.303 | 0.490 | 0.323 | 0.421 |

transfer learning for object detection tasks. As shown in figure 2, increasing the value of $\beta$ helps reduce model overconfidence and produces predictions that are less 'peaky' compared to label smoothing and hard label settings. Another interesting trend can be observed by changing the value of the $\beta$ parameter. As $\beta$ decreases in value, the overconfidence rate goes up along with it as shown in table 2. In case of transfer learning, we observe that decreasing $\beta$ causes the object localization performance to drop. Using objectness information helps our CNNs localize and detect objects better than the hard label baseline. Context dependence can be controlled using $\beta$.

## 5 CONCLUSION

This paper has addressed the problems of contextual bias and calibration using a novel approach called *adaptive* label smoothing. We show that bounding box information pertaining to objects can be used to compute a smoothing factor adaptively during training to improve the localization and calibration performance of CNNs. We use bounding box information for a portion of the ImageNet dataset (Russakovsky et al., 2015) to train different classifiers. We show that our approach can be used to train CNNs that are calibrated and have better localization performance on the challenging MS-COCO dataset (Lin et al., 2014) after fine-tuning, compared to approaches that use hard labels or traditional label smoothing approaches. Our labels implicitly capture the object proportion within an image during training, a significantly more challenging task than training with hard labels. Our methods provide the lowest accuracy and an order of magnitude reduction in average confidence when presented with context only images. We are extending this work to out of distribution detection as well. With adaptive label smoothing, when no pertinent objects are present, every class is equally probable for a given image. We introduce adaptive label smoothing with the notion that safety-critical applications need CNNs that are trained not to be overconfident in their predictions. Our intention is for decision making systems (steering inputs to an autonomous vehicle for example) to not make decisions in a definite way when the models are not confident in their predictions. Our approach provides a more reliable measure of confidence compared to all baselines.

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

## A  APPENDIX

We provide more detailed results and discussions that were left out due to space constraints in the main paper.

### A.1  ILLUSTRATION

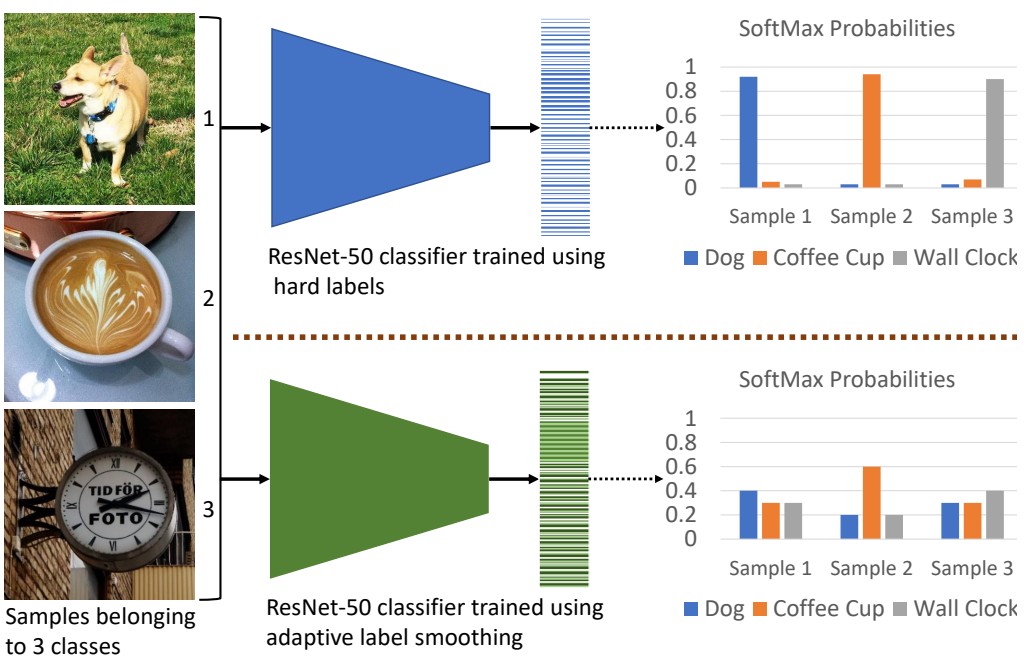

Figure 3: Hard-label and label-smoothing based approaches (top half of the figure) do not take into account the proportion of the object being classified. Our approach (bottom half) weights soft labels using the objectness measure to compute an *adaptive* smoothing factor during training, this helps produce peaks corresponding to object size during inference.

### A.2  DATASET

Our approach to create the different versions of ImageNet Russakovsky et al. (2015) to train our models are described in figure 4. We use the pixel means to mask all but one or all the objects using the same methodology as Anne Hendricks et al. (2018); Choi et al. (2019). We use the standard validation set along with ImageNet V2 Recht et al. (2019) without any changes to the images.

We also used a portion of the OpenImages Kuznetsova et al. (2020) dataset. More specifically, we used the object-detection version of the dataset, consisting of 600 classes and 1.7M images with bounding boxes. We selected a subset of these images and trained 5 classifiers.

In the case of OpenImages Kuznetsova et al. (2020), we use the object detection dataset consisting of 600 classes and 1.7M images with 14M bounding boxes. However, the 600 classes also include many parent nodes and as this can contribute to label confusion. We remove all parent node classes and use only the leaf node classes. The dataset has bounding boxes for only a subset of images for commonly occurring objects and we remove these classes as well. Finally, we follow the approach of Liu et al. (2020) and merge confusing classes. We end up with 480 classes and approximately 1.2M images. There are about 7 objects per image (average) in this subset and after applying the 'mask' method, we end up with approximately 6.8M images. Of these, about 1.3M images corresponded to the 'man' class and 'women' and 'windows' classes also had very high sample counts. We restrict the maximum number of images in a given class to around 50K and end up with roughly 2.2M images. We apply the same methodology to the val and test splits but we do not clip the sample counts per class.

Even after clipping the sample counts, the OpenImages dataset is very skewed compared to ImageNet as shown in figure 5, and we believe this imbalance makes OpenImages unsuitable for training good classifiers.

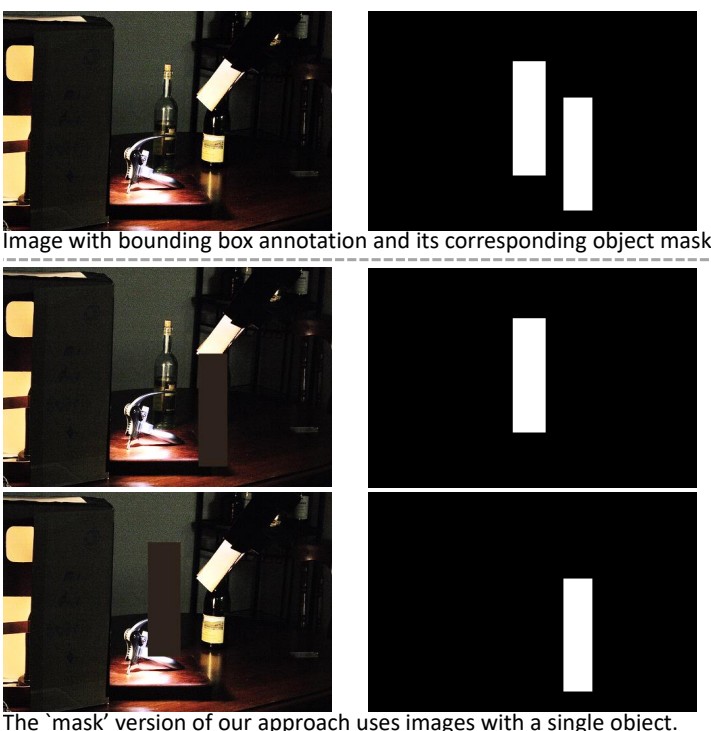

Image with bounding box annotation and its corresponding object mask.

The `mask' version of our approach uses images with a single object.

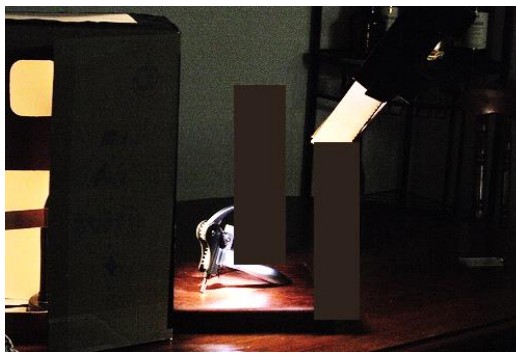

The `context' version of our approach uses images with all the objects masked out about 15% of the time during training. The label vector for such images (context only) is a vector of uniform distribution.

Figure 4: The first row of images in the left half of the figure are an example of the ImageNet dataset (N=0.474M) that have bounding box annotations. We match the images from the training set of ImageNet-1K dataset with the corresponding '.xml' files included in the ImageNet object detection dataset. We then create object masks for each of the images. When applying any scaling and cropping operation to training samples, we apply the same transformation to the corresponding object masks as well. By counting the number of white pixels, we can determine the object proportion post transformation. We describe the two other approaches in the figure, the 'mask' version of our approach has a single object (for images with multiple bounding box annotations) and this version has 0.528M samples. Our approach helps generate accurate labels during training and penalizes low-entropy (high-confidence) predictions for context-only images like the example on the right half of the figure.

Visualization of the count per each of the 1000 classes in the `mask' version of ImageNet used by our approach.

Visualization of the count per each of the 480 classes in the `mask' version of OpenImages used by our approach. Class `256 ' for example, has 40k images.

Figure 5: Top half of the figure shows the count per class for the ImageNet dataset, the highest number of images in a given class is '1349' and the lowest count is '190'. The distribution in this case is not as skewed as the OpenImages (bottom half) dataset. About 60 classes in our subset of the OpenImages dataset account for half the dataset. The maximum and minimum counts are, 55K and 28 respectively.

## A.3 HYPERPARAMETERS

We use standard data-augmentation strategies like random cropping, scaling, color jitter, etc., for all methods and train all our ImageNet models for 300 epochs starting with a learning rate of $0.1$ and decayed by $0.1$ at epochs 75, 150, and 225 using a batch size of 256. For a fair comparison with our ImageNet-'mask' based models, we matched the number of iterations and reduced the total epochs for our OpenImages classifiers. We trained all our OpenImages models for 72 epochs starting with a learning rate of $0.1$, and decayed by $0.1$ at epochs 18, 36, and 54 using a batch size of 256. We assume that this reduced number of epochs also contributed to poor localization for the transfer learning case.

## A.4 HARDWARE AND SOFTWARE

All our experiments were run on 'Dell C4130' nodes, equipped with 4 Nvidia V100 cards each. We used Docker to maintain the same set of libraries across multiple nodes. The host environment was running ubuntu 18.04 with cuda 10.2 installed. The docker environment used ubuntu 16.04 with cuda 9.0 and PyTorch 1.1 and Anaconda python 4.3. We will release all our code and pretrained models before the conference.

## A.5 RUNTIMES

Our *adaptive* label smoothing approach using the 'mask' version of ImageNet took approximately 74 hours and the hard label version took approximately 48 hours for 300 epochs. The object detection experiments took approximately 34 hours for 10 epochs.

## A.6 CLASS ACTIVATION MAPS

We provide more class activation maps to visualize the localization performance of baseline approaches, as well as our approaches in figures 6 8 and 7.

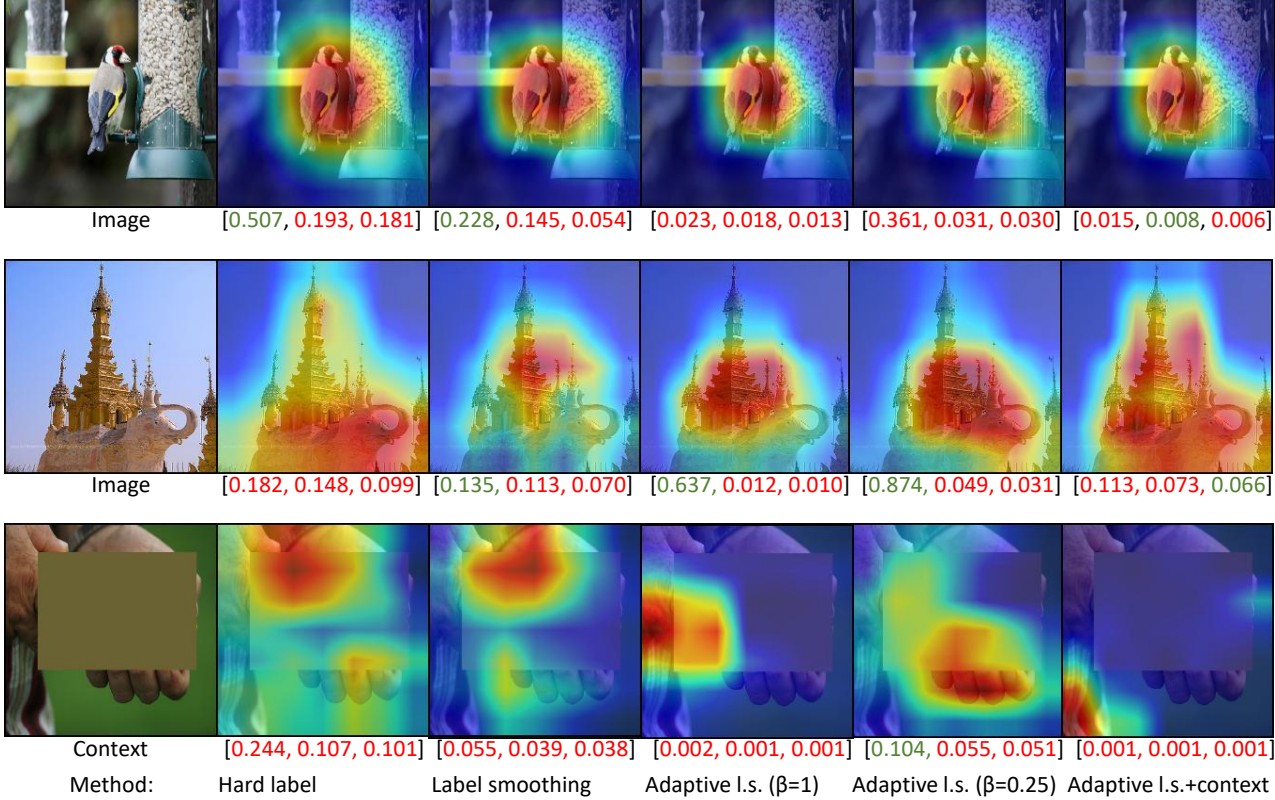

Figure 6: Examples of class activation maps (CAMs). These were obtained using the implementation of (Chattopadhay et al., 2018). The values under each CAM represent the top three probabilities, with green indicating the pertinent class and red indicating an incorrect prediction. Two columns on the left show results for baseline CNNs using hard labels and standard label smoothing. Our approach, *adaptive* label smoothing ('Adaptive l.s'), is illustrated in the three rightmost columns. Our technique produces high-entropy predictions on images without any objects and shows an improved localization performance.

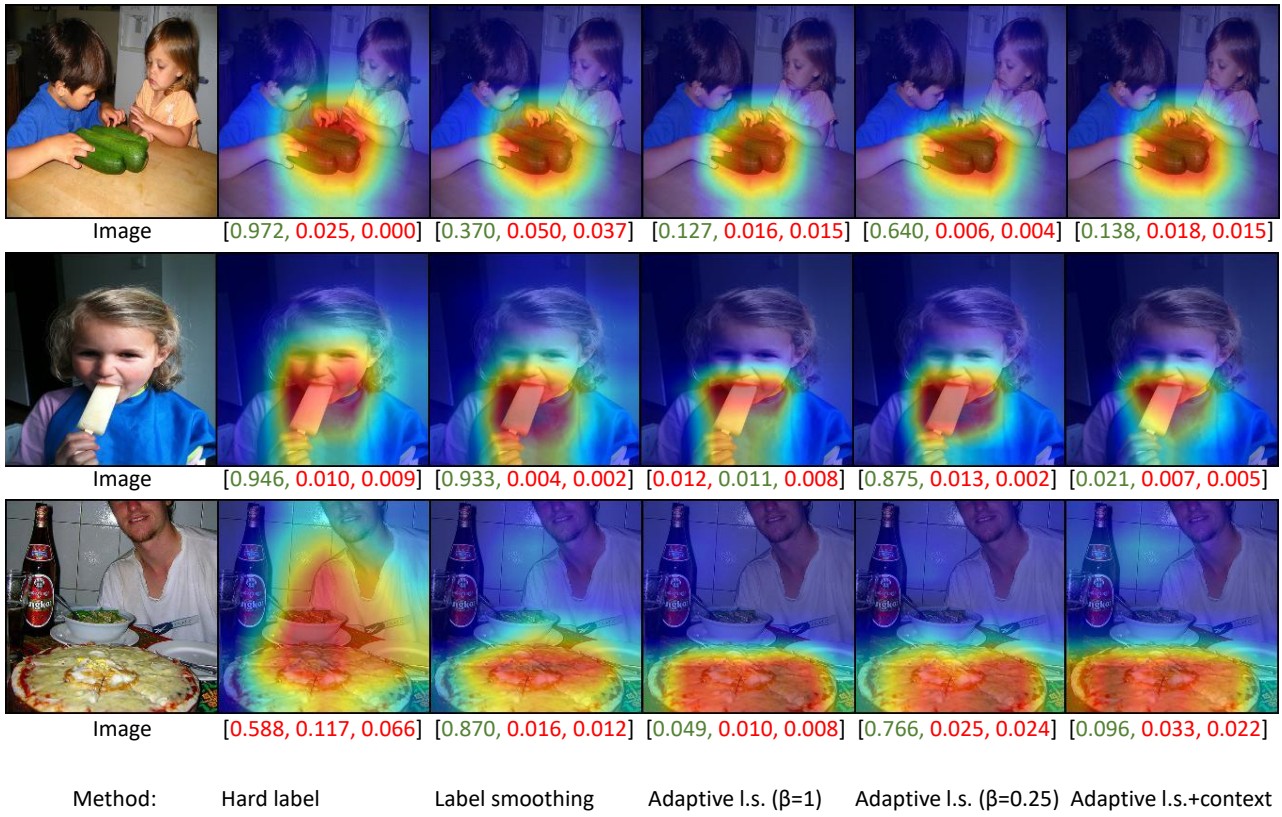

|  | Image | [0.972, 0.025, 0.000] | [0.370, 0.050, 0.037] | [0.127, 0.016, 0.015] | [0.640, 0.006, 0.004] | [0.138, 0.018, 0.015] |
| Image | [0.946, 0.010, 0.009] | [0.933, 0.004, 0.002] | [0.012, 0.011, 0.008] | [0.875, 0.013, 0.002] | [0.021, 0.007, 0.005] |
| Image | [0.588, 0.117, 0.066] | [0.870, 0.016, 0.012] | [0.049, 0.010, 0.008] | [0.766, 0.025, 0.024] | [0.096, 0.033, 0.022] |
| Method: | Hard label | Label smoothing | Adaptive l.s. (β=1) | Adaptive l.s. (β=0.25) | Adaptive l.s.+context |

Figure 7: Examples of class activation maps (CAMs). These were obtained using the implementation of (Chattopadhay et al., 2018). The second and third columns from the left show results for baseline CNNs using hard labels and standard label smoothing. Our approach, *adaptive* label smoothing ('Adaptive l.s'), is illustrated in the three rightmost columns. Our technique produces high-entropy predictions and shows an improved localization performance. The values under each CAM represent the top three probabilities, with green indicating the pertinent class and red indicating an incorrect prediction.

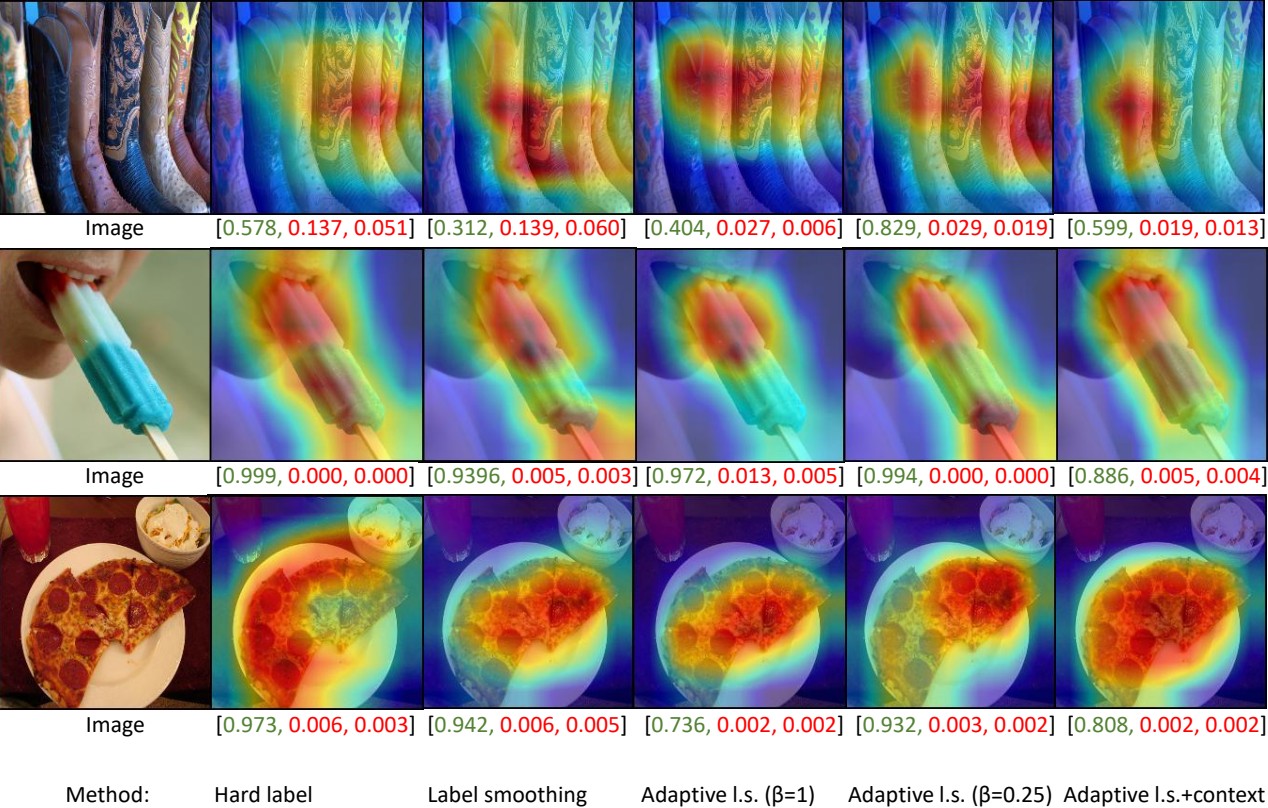

Figure 8: More examples of class activation maps (CAMs). These were obtained using the implementation of (Chattopadhay et al., 2018). The second and third columns from the left show results for baseline CNNs using hard labels and standard label smoothing. Our approach, *adaptive* label smoothing ('Adaptive l.s'), is illustrated in the three rightmost columns. Our technique produces high-entropy predictions and shows an improved localization performance. The values under each CAM represent the top three probabilities, with green indicating the pertinent class and red indicating an incorrect prediction.

## A.7 TABLES

We provide detailed calibration metrics for ImageNet and OpenImages classifiers in tables 5 and 6 respectively.Average confidence of a model describes the mean confidence of a model. As our model predictions are grounded in the spatial size of the object, our average confidence values on 'V1' and 'V2' are $0.48$ and $0.39$, respectively; in the case of hard labels the values are $0.77$ and $0.69$, respectively. We also provide AP (average precision) measures for different object sizes in table 7.

Table 5: Classification and calibration results with ImageNet. For a detailed explanation of the metrics please refer to section 4.2 in the main paper. 'A.conf', 'O.conf' and 'U.conf' refer to average confidence, overconfidence, and underconfidence scores. We provide ECE values for 100 bins and 15 bins mean scores along with their standard deviation (std).

| Method | Val. Set | Train (N) | Acc. mean | Log-loss mean | ECE 100 mean | ECE 15 mean | MCE mean | O.conf mean | U. conf mean | A. conf mean |
|---|---|---|---|---|---|---|---|---|---|---|
| Hard Label | V1 | 1.28M | 0.769 | 0.963 | 0.062 | 0.045 | 0.284 | 0.582 | 0.100 | 0.826 |
| Hard Label | V2 | 1.28M | 0.647 | 1.643 | 0.131 | 0.099 | 0.664 | 0.538 | 0.131 | 0.752 |
| CutMix | V1 | 1.28M | 0.788 | 0.882 | 0.035 | 0.022 | 0.267 | 0.520 | 0.162 | 0.770 |
| CutMix | V2 | 1.28M | 0.661 | 1.499 | 0.094 | 0.051 | 0.817 | 0.485 | 0.192 | 0.699 |
| RICAP | V1 | 1.28M | 0.782 | 0.896 | 0.032 | 0.021 | 0.284 | 0.553 | 0.131 | 0.800 |
| RICAP | V2 | 1.28M | 0.663 | 1.533 | 0.108 | 0.072 | 0.697 | 0.516 | 0.165 | 0.728 |
| | | | | | | | | | | |
| Hard Label | V1 | 0.474M | 0.669 | 1.568 | 0.104 | 0.093 | 0.347 | 0.558 | 0.123 | 0.771 |
| Hard Label | V2 | 0.474M | 0.543 | 2.365 | 0.171 | 0.148 | 0.759 | 0.520 | 0.154 | 0.697 |
| CutMix | V1 | 0.474M | 0.689 | 1.368 | 0.032 | 0.017 | 0.167 | 0.456 | 0.209 | 0.687 |
| CutMix | V2 | 0.474M | 0.577 | 2.021 | 0.100 | 0.050 | 0.517 | 0.421 | 0.248 | 0.612 |
| Label Smoothing | V1 | 0.474M | 0.691 | 1.428 | 0.055 | 0.051 | 0.354 | 0.401 | 0.248 | 0.643 |
| Label Smoothing | V2 | 0.474M | 0.558 | 2.107 | 0.102 | 0.047 | 0.512 | 0.368 | 0.283 | 0.563 |
| A. L.S. | V1 | 0.474M | 0.655 | 2.121 | 0.191 | 0.186 | 0.461 | 0.255 | 0.401 | 0.480 |
| A. L.S. | V2 | 0.474M | 0.532 | 2.839 | 0.185 | 0.158 | 0.661 | 0.217 | 0.441 | 0.399 |
| | | | | | | | | | | |
| Hard Label (mask) | V1 | 0.528M | 0.680 | 1.451 | 0.088 | 0.076 | 0.259 | 0.549 | 0.132 | 0.766 |
| Hard Label (mask) | V2 | 0.528M | 0.559 | 2.194 | 0.155 | 0.127 | 0.686 | 0.507 | 0.163 | 0.691 |
| CutMix (mask) | V1 | 0.528M | 0.698 | 1.326 | 0.032 | 0.020 | 0.249 | 0.477 | 0.197 | 0.704 |
| CutMix (mask) | V2 | 0.528M | 0.576 | 1.999 | 0.110 | 0.067 | 0.614 | 0.449 | 0.228 | 0.635 |
| Label Smoothing (mask) | V1 | 0.528M | 0.687 | 1.447 | 0.051 | 0.046 | 0.430 | 0.407 | 0.244 | 0.647 |
| Label Smoothing (mask) | V2 | 0.528M | 0.563 | 2.135 | 0.108 | 0.048 | 0.524 | 0.374 | 0.281 | 0.568 |
| A. L.S. (mask) | V1 | 0.528M | 0.648 | 2.176 | 0.186 | 0.182 | 0.463 | 0.246 | 0.396 | 0.478 |
| A. L.S. (mask) | V2 | 0.528M | 0.528 | 2.914 | 0.185 | 0.160 | 0.687 | 0.209 | 0.441 | 0.394 |
| A. L.S. (mask) (beta =0.75) | V1 | 0.528M | 0.681 | 1.759 | 0.146 | 0.142 | 0.377 | 0.319 | 0.337 | 0.553 |
| A. L.S. (mask) (beta =0.75) | V2 | 0.528M | 0.556 | 2.478 | 0.146 | 0.113 | 0.572 | 0.274 | 0.375 | 0.469 |
| A. L.S. (mask) (beta =0.25) | V1 | 0.528M | 0.684 | 1.479 | 0.059 | 0.052 | 0.244 | 0.402 | 0.244 | 0.645 |
| A. L.S. (mask) (beta =0.25) | V2 | 0.528M | 0.561 | 2.191 | 0.109 | 0.059 | 0.627 | 0.369 | 0.285 | 0.563 |
| A. L.S. + Context (mask) | V1 | 0.528M | 0.637 | 2.197 | 0.174 | 0.169 | 0.431 | 0.251 | 0.390 | 0.480 |
| A. L.S. + Context (mask) | V2 | 0.528M | 0.515 | 2.954 | 0.177 | 0.147 | 0.682 | 0.221 | 0.437 | 0.397 |
| A. L.S. + CutMix (mask) | V1 | 0.528M | 0.442 | 4.569 | 0.349 | 0.332 | 0.559 | 0.047 | 0.843 | 0.095 |
| A. L.S. + CutMix (mask) | V2 | 0.528M | 0.346 | 4.952 | 0.292 | 0.265 | 0.902 | 0.049 | 0.851 | 0.083 |

Table 6: Classification and calibration results with OpenImages. For a detailed explanation of the metrics please refer to section 4.2 in the main paper. 'A.conf', 'O.conf' and 'U.conf' refer to average confidence, overconfidence, and underconfidence scores. We provide ECE values for 100 bins and 15 bins mean scores along with their standard deviation (std).

| Method | Val./Test size | Val. Set | Acc. mean | Log-loss mean | ECE 100 mean | ECE 15 mean | MCE mean | O.conf mean | U. conf mean | A. conf mean |
|---|---|---|---|---|---|---|---|---|---|---|
| Hard Label (mask) | 105978 | Val | 0.552 | 1.519 | 0.089 | 0.080 | 0.280 | 0.476 | 0.235 | 0.636 |
| Hard Label (mask) | 325098 | Test | 0.549 | 1.522 | 0.089 | 0.083 | 0.262 | 0.479 | 0.238 | 0.634 |
| Label Smoothing (mask) | 105978 | Val | 0.554 | 1.573 | 0.044 | 0.032 | 0.220 | 0.410 | 0.312 | 0.564 |
| Label Smoothing (mask) | 325098 | Test | 0.550 | 1.577 | 0.033 | 0.029 | 0.196 | 0.408 | 0.315 | 0.561 |
| A. L.S. (mask) | 105978 | Val | 0.392 | 4.725 | 0.389 | 0.372 | 0.779 | 0.032 | 0.908 | 0.055 |
| A. L.S. (mask) | 325098 | Test | 0.388 | 4.749 | 0.346 | 0.328 | 0.579 | 0.031 | 0.912 | 0.053 |
| A. L.S. (mask) + Context | 105978 | Val | 0.383 | 4.049 | 0.219 | 0.203 | 0.464 | 0.092 | 0.626 | 0.200 |
| A. L.S. (mask) + Context | 325098 | Test | 0.371 | 4.092 | 0.193 | 0.178 | 0.415 | 0.089 | 0.624 | 0.195 |
| A. L.S. (mask) (beta =0.25) | 105978 | Val | 0.556 | 1.667 | 0.058 | 0.051 | 0.226 | 0.371 | 0.362 | 0.519 |
| A. L.S. (mask) (beta =0.25) | 325098 | Test | 0.554 | 1.670 | 0.052 | 0.049 | 0.127 | 0.370 | 0.364 | 0.517 |

Table 7: Fine-tuning on COCO using FRCNN for object detection. For a detailed explanation of the results please refer to section 4.3 in the main paper. AP refers to average precision and AR refers to average recall at the specified Intersection over union (IoU) level. We also provide AP values for small, medium, and large objects using 'S', 'M', and 'L' respectively.

| Method | Pre-train dataset | Pre-train size | AP 0.5:0.95 | AP 0.5 | AP 0.75 | AP (S) 0.5:0.95 | AP (M) 0.5:0.95 | AP (L) 0.5:0.95 |
|---|---|---|---|---|---|---|---|---|
| Hard Label | ImageNet | 1.28M | 0.323 | 0.519 | 0.345 | 0.136 | 0.367 | 0.481 |
| CutMix | ImageNet | 1.28M | 0.329 | 0.528 | 0.353 | 0.139 | 0.376 | 0.490 |
| RICAP | ImageNet | 1.28M | 0.331 | 0.528 | 0.354 | 0.138 | 0.376 | 0.493 |
| Hard Label | ImageNet | 0.474M | 0.290 | 0.479 | 0.309 | 0.112 | 0.325 | 0.437 |
| Adaptive L.S. | ImageNet | 0.474M | 0.311 | 0.501 | 0.332 | 0.119 | 0.352 | 0.470 |
| Hard Label (mask) | ImageNet | 0.528M | 0.290 | 0.482 | 0.307 | 0.114 | 0.329 | 0.435 |
| CutMix (mask) | ImageNet | 0.528M | 0.312 | 0.509 | 0.329 | 0.125 | 0.353 | 0.470 |
| Label Smoothing (mask) | ImageNet | 0.528M | 0.304 | 0.500 | 0.324 | 0.122 | 0.346 | 0.455 |
| Adaptive L.S. (mask) | ImageNet | 0.528M | 0.311 | 0.501 | 0.333 | 0.124 | 0.351 | 0.477 |
| Adaptive L.S. (mask) (beta =0.75) | ImageNet | 0.528M | 0.309 | 0.498 | 0.331 | 0.123 | 0.348 | 0.467 |
| Adaptive L.S. (mask) (beta =0.25) | ImageNet | 0.528M | 0.298 | 0.492 | 0.315 | 0.122 | 0.340 | 0.449 |
| Adaptive L.S. + Context (mask) | ImageNet | 0.528M | 0.303 | 0.490 | 0.323 | 0.115 | 0.339 | 0.465 |
| Adaptive L.S. + CutMix (mask) | ImageNet | 0.528M | 0.273 | 0.449 | 0.289 | 0.098 | 0.300 | 0.423 |
| Hard Label (mask) | OpenImages | 1.20M | 0.295 | 0.484 | 0.313 | 0.115 | 0.330 | 0.453 |
| Label Smoothing (mask) | OpenImages | 1.20M | 0.301 | 0.493 | 0.320 | 0.119 | 0.339 | 0.457 |
| Adaptive L.S. (mask) | OpenImages | 1.20M | 0.243 | 0.415 | 0.250 | 0.083 | 0.263 | 0.376 |
| Adaptive L.S. + Context (mask) | OpenImages | 1.20M | 0.289 | 0.471 | 0.308 | 0.111 | 0.321 | 0.448 |
| Adaptive L.S. (mask) (beta =0.25) | OpenImages | 1.20M | 0.304 | 0.494 | 0.324 | 0.118 | 0.340 | 0.462 |

