# OpenReview forum: "One Size Doesn't Fit All: Adaptive Label Smoothing"
_ICLR.cc/2021/Conference — Reject_

### Official Review · AnonReviewer3 · 2020-10-17
**Nice idea but the experiments are not meeting the expectations**

**Rating:** 4
**Confidence:** 4

**Review:**

**Summary**
Many modern classifiers are trained with class labels - it's natural because you train the model with precisely the target label you want the model to produce! But let's think out of the box and introduce stronger forms of supervision - e.g. bounding boxes for the object of interest. What benefits will they bring? The paper is arguing that the added strong supervision (boxes on top of ImageNet dataset) helps image classifiers reduce their over-confidence. This is done by training the classifier with the **adaptive label smoothing**, where each CNN input is assessed according to the actual proportion of the foreground object and the target vector is label smoothed to reflect the proportion.

**Pros**
The paper is one of those papers which dig into the overlooked bits in the widespread practice. Given that the point is valid and solidly tested under extensive experiments, the paper is likely to leave a strong mark in the research community, providing researchers and practitioners chances to re-think about their habits and hidden assumptions. This paper has great potential to guide the researcher in such a direction. I believe we can also learn from this paper potentials to use strong supervision for improving not only the main task performance (e.g. classification) but also guide "how" the models should achieve the recognition - e.g. by improving the input attribution (explainability), robustness (adversarial and natural), bias (e.g. texture or bg bias), and uncertainty (as done in this paper).

**Cons**
It is a little unfortunate that the paper falls short in terms of presentation and experimental depth. It is not fully convincing yet that the proposed solution is working, given the width and depth of experiments. I would highly suggest the authors re-structure the paper and add substantially more experimental validations for the next paper. Below are more specific comments on the weaknesses.

1. **The proposed adaptive label smoothing actually worsens many of the key performance metrics.**
In Table 1, we observe that compared to "hard label" (0.669 top-1 accuracy), "A. L. S." (or adaptive label smoothing) has only 0.655 top-1 accuracy on ImageNet validation set. It is a bit difficult to swallow the result, given that ALS is **requiring more annotation budgets** (bounding boxes) to prepare the training set. It is not only the main task where ALS is falling short. Also on calibration metrics (ECE and MCE), ALS is doing **worse** than the baseline, with higher ECE and MCE measures (lower is better). The authors argue that ALS successfully decreases the overconfidence, which is true indeed, but this is only half of the story! By decreasing the overall confidence, ALS turns out to further decrease confidence in correct prediction (underconfidence scores). Table 1 seems to suggest that ALS is actually not working.

2. **Given the breadth and ramifications of the claim (see pros above), the set of experiments seems limited (only ImageNet + ResNet50 + COCO transfer learning).**
Given that the issue 1 above is resolved, I would still suggest doing more experiments on more datasets (e.g. OpenImages?) and more architectures (e.g. Other ResNet or EfficientNet families) to ensure that ALS is actually working, independent of the specific dataset-architecture pair.

3. **Structure the paper better.**
The paper spends multiple paragraphs repeating points that are already made (sections 1,2) and spends so little on important implementation details and evaluation setups (section 4). There are seven paragraphs in section 1 - please aim to make it four paragraphs. This will include moving the experimental analysis on the confidence scores to section 4 (introduction is not a good place to already talk about numbers!). Try to shorten the six paragraphs in section 2 into three paragraphs.
The paper will now be only 6.5-7 pages long. With the remaining pages, describe the following in greater details:
- The different splits of ImageNet-1K training set: please give them names & use the designated names in Tables 1&2.
- Write down the precise implementation details and the definitions of evaluation metrics in 4.2 (e.g. ECE, MCE, under/overconfidence) instead of referring readers to the previous papers. Please indicate whether higher or lower values are better (also in the tables).
- When discussing the results, please talk about **all** the results for all evaluation metrics, instead of just overconfidence, as done in section 4.2. Please also talk with numbers, instead of just saying e.g. "these results have a low overconfidence score".

4. **Nits**
* When I checked last time, ImageNet training set had **42%** images annotated with bounding boxes. Is **38%** (section 4.1) correct?
* (Lin et al., 2014) --> MS COCO (Lin et al., 2014)
* described in -2 --> described in table 2
* allows the CNNs to cheat: I find it difficult to agree with that statement. There are many papers saying the use of context has helped a lot in their application scenarios. Please tone it down.

5. **Final idea to throw**
Did you consider using unsupervised objectness methods - like Edgebox, Selective search, MCG object proposals? Or saliency detection methods (https://paperswithcode.com/task/salient-object-detection)? Being able to use them will replace your need for bounding box supervision.

**Key reasons for the rating**

While the paper has and triggers many great ideas for researchers in this field, the experimental results are not supporting the main idea and claims. The presentation is quite a bit of an issue too. My suggestion is to substantially improve and scale-up the experiments for the next conference.

**Response to the last comments made by the authors**
The referenced paper (https://arxiv.org/pdf/1906.04933.pdf ) is not answering the question I have. Nor does section 4.2.

My question is: does ALS really improve the quality of uncertainty?

The paper's answer seems to be no.

Look at Tables 1, 2, 3, and 4 in the revised paper. It is great that ALS achieves a lower O.conf (overconfidence), meaning that for wrong predictions, ALS helps to produce lower confidence values. However, this comes at the cost of higher U.conf (underconfidence), meaning that even for correct predictions, ALS makes the model produce low confidence values. A confidence measure that always produces a low value, regardless of whether the prediction was correct or not, is not useful.

The authors may say "look at Table 1 - our uncertainty measure produces lower scores for images with objects removed". While this is true and it is a good signal, object removal is only a particular case for introducing uncertainty in an image. Eventually, I believe a good uncertainty measure should first of all produce a good ranking of test images such that the correctly predicted images are ranked first (this paper does not quantify this). Then comes the question of calibration (like ECE and MCE). Then comes the evaluations with specific uncertainty scenarios (like OOD images, object removal, or other controls on image uncertainty).

---

> ### Author Response · Authors · 2020-11-25
> **We are grateful for AnonReviewer3’s time, comprehensive comments and suggestions for improvement in the review of this paper. We appreciate the recognition of potential implications that our method could have in the research community. Below we address the concerns as outlined by R3 to improve the quality and presentation of our paper.**
>
> “The proposed adaptive label smoothing actually, ….. ALS is actually not working.”: Our accuracy is lower as we discourage our model from relying on context to make correct predictions. CutMix is able to report an accuracy of 9.2% even when no objects are present! Traditional approaches might produce excellent results on a given dataset, but such approaches suffer when deployed for real-world applications.The benefit of the method is that, under uncertain conditions (real-world conditions), we are able to produce an order of magnitude reduction in average confidence when no pertinent objects are present in a given image. On a given dataset, context helps traditional classifiers make correct predictions, but for real-world applications, such reliance can result in harmful effects (as in the case of autonomous driving). Our approach adds additional penalty terms to the loss function as we explain in the paper. ECE and MCE help with quantifying the discrepancy between a model's predictions and its accuracy. Our predictions are grounded in objectness scores and that is the reason why we are more accurate than we are confident compared to other baselines. The underconfidence is higher also because we are not trying to predict with the maximum possible confidence of 1, but we predict with a confidence proportional to the objectness score. Underconfidence has no effect on accuracy as argmax of the vector is used to assign class labels. We have added additional discussion to the paper explaining how ALS is actually matching objectness of the imagenet validation samples better.
>
> “experiments on more datasets (e.g. OpenImages?) and more architectures”: We have added ResNet101 and Openimages results. These were left out of the original submission due to space considerations. The main point of our paper is to demonstrate that our approach (adaptive label smoothing) is effective at addressing the problem of overconfident predictions. We believe that ResNet-50 is more than adequate to illustrate this basic principle. We have added experiments with varying amount of context dependence to understand the effect. Please see section A.6, which shows that our approach produces extremely low confidence values when presented with background images.
>
> "Structure the paper better.": These suggestions are very helpful and we have done our best to accommodate. We will further revise the submission before the camera-ready deadline. Thank you.
>
> “When I checked last time, ImageNet training set had 42% images annotated with bounding boxes. Is 38% (section 4.1) correct?”: We matched the .xml annotation files based on the filenames in the training set. It is possible that some annotations might have been filtered out or got removed if they corresponded to the val set images.
>
> “allows the CNNs to cheat: I find it difficult to agree with that statement. There are many papers saying the use of context has helped a lot in their application scenarios. Please tone it down.”: We have changed the language, and we apologize for the negative connotation. Context is absolutely important. However, relying only on context to make correct predictions is not a good way to solve problems. We wanted to highlight the problem with some patch based methods (CutMix, RICAP). Even though they produce state of the art results on a given dataset, they are not applicable for real world applications or safety-critical applications (predicting a whale based on an underwater image containing no whale, for example).
>
> “Final idea to throw Did you consider using unsupervised objectness methods - like Edgebox, Selective search, MCG object proposals? Or saliency detection methods (https://paperswithcode.com/task/salient-object-detection)? Being able to use them will replace your need for bounding box supervision.”: There are many ways to compute objectness, but our goal is not to understand the differences in the objectness approaches. Instead, we addressed the question: given a way to compute objectness, can providing accurate labels reduce contextual bias?
>
> “While the paper has and triggers many great ideas for researchers in this field, the experimental results are not supporting the main idea and claims.”: The results are supportive, we do have state of the art results under total uncertainty. Validation sets are guaranteed to have objects encountered in the training set, but real-world has no such guarantees.
>
> We have updated the submission based on your feedback and suggestions. We hope AnonReviewer3 will take these changes into account. We welcome additional suggestions for our camera-ready version as well.

---

> > ### Comment · AnonReviewer3 · 2020-11-25
> > **Thank you for the response. But the response does not fully address the concern.**
> >
> > Thank you for the detailed response and the paper updates with the new OpenImages and ResNet101 results.
> >
> > Unfortunately, I do not think the response and the revision are addressing the gist of the concerns raised by me and some other reviewers - that the benefit of the method is not quantified. It is difficult to conclude from the results tables 1, 2, 3, and 4 that I should definitely use the proposed ALS over the vanilla baseline. In general, it is difficult to say that the uncertainty measures predict the correctness of the predictions well (ECE and MCE scores are not improving). Nor are the classification performances definitely better. The authors have defended this by saying that the uncertainty estimates are decreased in general. This, however, does not necessarily mean the uncertainty measures are *better*. A good uncertainty measure should discern correct predictions from wrong predictions well. At the current state, ALS simply produces lower uncertainty measures for both correct and wrong predictions (see overconfidence and underconfidence evaluations). We need *both* low overconfidence and low underconfidence. I am repeating the argument several times to make sure that the authors get the point. If we only care about lowering the overconfidence, then the simplest solution is to set the confidence level to zero for every test image - which obviously is not a good uncertainty measure.
> >
> > > we are able to produce an order of magnitude reduction in average confidence when no pertinent objects are present in a given image.
> >
> > Well this is true, but again ALS is producing lower confidence scores even for images with foreground objects. The paper needs to quantify that the ALS produces lower confidence scores for images without FG objects and higher scores for images with FG objects. The qualitative samples as given in Figure 3 don't work for me, I'm afraid. There should be a quantifiable evidence that ALS truly improves the quality of confidence measures.
> >
> > I truly appreciate the efforts of the authors, but it is regrettable that they have missed the point. It would have been better if the authors could have communicated with the reviewers earlier on. I find it difficult to raise my score based on the authors' rebuttal this time.

---

> > > ### Author Response · Authors · 2020-11-25
> > > **response**
> > >
> > > https://arxiv.org/pdf/1906.04933.pdf
> > > Please read S1 of the paper to understand why over and underconfidence are not independent of each other.
> > > "Hence, aleatoric and epistemic uncertainty contained in the data distribution influence the over-and underconfidence of a classifier."
> > >
> > > Last paragraph of section 4.2 explains why ALS is working, we match objectness per sample much better compared to the baseline.

---

### Official Review · AnonReviewer1 · 2020-10-28
**An adaptive label smoothing approach to learn localized visual representations**

**Rating:** 4
**Confidence:** 4

**Review:**

**Summary:**

This paper proposed an approach to learn localized visual representation by using bounding boxes annotations. It uses a label smoothing approach where the smoothing parameter is computed per example based on the proportion of the object in the image. The approach is tested on image classification and object detection.

**Reasons for score:**

I do not think the technical contribution is strong enough for ICLR. The idea of computing the smoothing parameter per example is interesting but it is not enough. I also think that the comparison to some similar works is missing.

**Pros:**

- The idea of learning localized representation is interesting because it allows better performance.
- I like the idea to compute the smoothing parameter per example where the value is the proportion of the object in the image.
- I like the analysis to quantify the context dependence.
- The paper is easy to read.


**Cons:**

Overall, I think the technical contribution of this paper is not enough for ICLR. The method section is very small and is based on the label smoothing mechanism. The main novelty is that the smoothing parameter is computed per example and its value is the proportion of the object in the image.

The paper only focus on the crop-based methods but ignore thee weakly-supervised learning approaches [2, 3, 4, 5]. These papers propose models to automatically find the objects in the images and learn some localized representations. The authors should compare with this types of approaches because they tackle the same problem.

The idea of using bounding boxes to train deep models is not new. For example, [1] uses bounding boxes. Using bounding boxes allows to learn better representation but these annotations are costly to get. It is easier to collect image classification labels than bounding boxes annotations. It is challenging to build a dataset of 1M images annotated with bounding boxes so it limits the potential impact of the proposed approach.

Overall, I like the analysis to quantify the context dependence. I just wonder how the mean pixel values and the shape of the box contain some information. For example, is it possible to predict the class of an object by using the shape of the bounding box. I think it can be interesting to verify it.

It is difficult to read the tables 1 and 2. The authors should improve the presentation of these tables.

The authors should comment if it is possible to generalize this approach to a multi-label dataset like MS COCO or OpenImages. Natural images are usually multi-labels because the world is a composition of objects.

In section 4.3, the authors wrote they trained models on OpenImages but they did not present the results.

[1] Oquab ., Bottou ., Laptev ., Sivic . Learning and Transferring Mid-Level Image Representations using Convolutional Neural Networks. In CVPR, 2014.
[2] Oquab ., Bottou ., Laptev ., Sivic . Is Object Localization for Free? - Weakly-Supervised Learning With Convolutional Neural Networks. In CVPR, 2015.
[3] Sun C., Paluri M., Collobert R., Nevatia R., Bourdev L. ProNet: Learning to Propose Object-specific Boxes for Cascaded Neural Networks. In CVPR, 2016.
[4] Zhou B., Khosla A., Lapedriza A., Oliva A., Torralba A. Learning Deep Features for Discriminative Localization. In CVPR, 2016.
[5] Durand ., Mordan ., Thome ., Cord . WILDCAT: Weakly Supervised Learning of Deep ConvNets for Image Classification, Pointwise Localization and Segmentation. In The IEEE Conference on Computer Vision and Pattern Recognition (CVPR), 2017.

---

> ### Author Response · Authors · 2020-11-25
> **We thank AnonReviewer1 for their efforts in this review and appreciate the positive comments in regard to adaptive smoothing factors. Additionally, we address R1’s suggestions for improvements.**
>
> “visual representation by using bounding boxes annotations.”: We think the reviewer might have focussed on effect rather than cause. Our goal is not to learn better localization, our goal is to supply accurate labels during training and penalize high confidence prediction when no evidence/object is present in the input. We are interested in CNN calibration and better localization is an emergent behavior. Please see section A.6, our approach produces extremely low confidence values when presented with background images.
>
> “I do not think the technical contribution is strong ,.....similar works is missing.”: Please the updated section 4.2 describing our results and the significance. Our approach adds additional penalty terms to the loss function as we explain in the updated paper. We thoroughly demonstrate the effect of context dependence using ablation studies. We used the latest baselines (label smoothing and CutMix) to compare our approach. ICLR is the best venue to discuss the different representations we learn using our approach. We also discuss the main issues with traditional cross entropy loss. Predicting with a high confidence even when there is no evidence makes traditional classifiers ill suited for real-world deployment.
>
> “The method section is very small and is based on the label smoothing mechanism”: We have expanded the method section and added discussion on cross entropy loss. And how our approach is different compared to label smoothing.
> “crop-based methods but ignore the weakly-supervised learning ”: We do not focus on crop based methods for object localization. Instead, we explore the problem of context dependence when wrong labels are supplied during training. We do not tackle object localization, but instead we use bounding boxes to compute objectness scores. As R3 and R2 pointed out, there are many ways to compute objectness, even in an unsupervised way. Weakly supervised object localization (WSOL) methods are finetuning ImageNet learnt representation, we are tackling classification with cross entropy loss smoothed using objectness score. We do not use GAP or new layers. Instead, we perform strict classification with soft labels instead of hard labels, using objectness to specify how object-rich the input is. We are interested in reducing classifier overconfidence. The better localization is a by-product, and not our main task. Our goal is not to produce attention maps to come up with better object proposals, but instead our goal is to supply accurate labels during training. Instead of supplying a binary value of 1 or 0, we supply a value based on the objectness score. WSOL methods can be used to provide gross objectness values for datasets where such information is not available.
> “Overall, I like the analysis to quantify the context dependence. I just wonder how the mean pixel values and the shape of the box contain some information. For example, is it possible to predict the class of an object by using the shape of the bounding box. I think it can be interesting to verify it.”: The reason behind using mean pixel values is to not skew the input distribution, especially when batch normalization is used. It would be interesting to quantify this bias as well.
>
> “It is difficult to read the tables 1 and 2”: We have streamlined the tables and moved the larger versions to the appendix.
>
> “generalize this approach to a multi-label dataset like MS COCO or OpenImages”: It should be able to generalize, but as we operate in a classification domain we are concerned with classifying one object at a time. The train split that we identify with 0.474M in table 5 has multiple objects present naturally and our results are consistent. More information is now available in the appendix. We combined sample mixing (CutMix) and our approach, but mixing samples from different contexts was not helpful.
>
> “The idea of using bounding boxes to train deep models is not new. For example, [1] uses bounding boxes. Using bounding boxes allows to learn better representation but these annotations are costly to get. It is easier to collect image classification labels than bounding boxes annotations. It is challenging to build a dataset of 1M images annotated with bounding boxes so it limits the potential impact of the proposed approach.”: Weakly supervised object localization (WSOL) methods can be used to provide gross objectness values for datasets where such information is not available. There are many ways to quantify objectness.
> “trained models on OpenImages .”: We left out OpenImages due to lack of space, but have now added classification and transfer learning results to the appendix.
>
> We have updated the submission and did our best to explain why our goal is not object localization. We hope AnonReviewer1 will take these changes into account. We welcome additional suggestions for our camera-ready version as well.

---

### Official Review · AnonReviewer2 · 2020-10-28
**Interesting idea for a relevant problem, but the experimental evaluation is limited**

**Rating:** 4
**Confidence:** 3

**Review:**

Summary:
This paper introduces the concept of adaptive label smoothing (ALS).  ALS uses bounding box annotations of objects to determine how much area in the image is covered by the object and uses this estimate to change the weight in the label smoothing operation, with the goal to encourage the classifier to focus on the object and not on the context around the object during classification. The experiments on ImageNet classification show an improved classifier calibration, while for detection, the improvements are marginal.

Pros:
- The paper is well written and easy to understand
- The problem of reducing the effect of background biases on a classifier's performance is important
- The classifier calibration on ImageNet is improved

Cons:
- The method requires additional supervision in terms of bounding box annotations. It would be interesting to study unsupervised methods for estimating the objectness, e.g., using unsupervised segmentation.
- The proposed method does not significantly improve over the baselines on object detection. I think the authors should discuss this in more detail.
- It would be great if the paper would also discuss alternative measures for adapting the smoothing factor.
- The overall contribution of this paper is rather limited. While the idea is certainly very interesting and the problem is important, the authors should give a more elaborate experimental evaluation of their approach on a variety of datasets and with a larger set of network architectures and for different visual recognition tasks to highlight that their strategy indeed is of general relevance.

----------------------------------------------------------------------------------------------------------
Post rebuttal:

Thank you for your response. After reading the other reviewers' comments and your responses, I think the paper is not yet ready for publication. All reviewers are concerned by the lack of a technical contribution and the limited benefit of the paper. While I appreciate the effort of the authors to answer our concerns, I think the paper needs a major revision that incorporates our shared concerns. Therefore, I retain my initial rating.

---

> ### Author Response · Authors · 2020-11-25
> **We appreciate AnonReviewer2’s time in the review of our paper and are pleased that R2 recognizes the importance of minimizing the contextual/background bias. We address concerns from R2 as follows:**
>
> “While for detection, the improvements are marginal.” and “The proposed method does not significantly improve over the baselines on object detection. I think the authors should discuss this in more detail.”: Our AP is 0.001 lower compared to CutMix (state of the art using ResNet50).
>
> “The method requires additional supervision in terms of bounding box annotations. It would be interesting to study unsupervised methods for estimating the objectness, e.g., using unsupervised segmentation.”: While objectness can be computed using other approaches (as pointed out by R3), we use bounding boxes as they are available for the ImageNet dataset to implement our adaptive label smoothing approach. We are not relying on object coordinates, but we use the information to compute foreground vs. background ratio.
>
> “It would be great if the paper would also discuss alternative measures for adapting the smoothing factor.”: The intuition is to ground the prediction in evidence, and this evidence could be calculated in a way that is dependent on the task and labels being supplied. Objectness can be computed using edge or basic visual cues as well. Other applications like speech recognition can use our approach when overlaying ambient noise on clean speech samples to augment data. The smoothing factor would indicate the strength of the signal compared to noise in the context of speech recognition.
>
> “The overall contribution of this paper is rather limited. While the idea is certainly very interesting and the problem is important, the authors should give a more elaborate experimental evaluation of their approach on a variety of datasets and with a larger set of network architectures and for different visual recognition tasks to highlight that their strategy indeed is of general relevance.”: The benefit of the method is that, under uncertain (real-world) conditions we are able to produce an order of magnitude reduction in average confidence when no pertinent objects are present in a given image. Please the updated section 4.2 describing our results and the significance. Our approach adds additional penalty terms to the loss function as we explain in the paper. The main point of our paper is to demonstrate that our approach (adaptive label smoothing) is effective at addressing the problem of overconfident predictions. We believe that ResNet-50 is more than adequate to illustrate this basic principle. We have now added ResNet-101, results using OpenImages, and ablation results by varying the amount of context dependence. Classification, class activation maps and object detection are basic but important visual tasks. We conduct experiments under total occlusion as well. Our confidences/predictions are based on the size of an object and we are more accurate than we are confident compared to all baselines. Please see section A.6, our approach produces extremely low confidence values when presented with background images.
>
> We have updated the submission and hope AnonReviewer2 will take these changes into account. We welcome additional suggestions for our camera-ready version as well.

---

### Official Review · AnonReviewer4 · 2020-11-01
**Simple idea but results are not convincing**

**Rating:** 4
**Confidence:** 4

**Review:**

This paper is about a study of calibration of deep neural networks in the image classification task, in particular to the possibility of using label smoothing to make the network emit predictions that are more accurate in terms of confidence when few or no pixels of an object are present in the input. The idea is to adjust the entropy of the labels in proportion to the amount of objectness of an image i.e. the amount of pixels related to the object. Experiments shows that the adjusted labels reduce the overconfidence of a classification network.

Strengths:
+ the paper is easy to read and presented well, beside some small issues that can be fixed easily.
+ related works are comprehensive and the method is really easy, which is a plus.

Weaknesses:
- experiments and results show limited benefit from the method. It seems that the only impact is in the object classification task where the overconfidence is reduce at the expense of the underconfidence, which means that there is a tradeoff in the confidence of trained classifier. It is not clear when it should be used and why. Moreover, the object detection results show similar performance of AP, even inferior of CutMix, without a proper measurement of confidence in that case.
- the method needs the amount of percentage of pixels of the object to adjust the labels. In practise, it needs the bounding boxes of the objects. This naturally suggests that the task should be evaluated in terms of object detection task. Results in this regards show that after finetuning there is minimal or negative impact (e.g. beta <<), rendering the method useless to my understanding.

In particular:
- OpenImages dataset is mentioned in 4.3, but COCO is used. COCO is not described anywhere.
- It is not clear the experimental setting when there are multiple annotated objects (sec 4.1). There is mention that 54k more images are derived but it is not clear how.
- It would be interesting to compare the method with a baseline neural network that not only emits the label of an object and thus we need to compute the entropy of the answer, but also an additional output with the confidence of the score.
- Discussion in 4.2 does not discuss the other metrics ECE, U.conf, MCE enough to give some insights about the results. In particular the U.conf gets higher with the method. What is happening and how does this affects the confidence?
- It is not clear in table 2 what are the two experiments with 0.474M pre train N and what information they give.
- Sec 4.4 mentions table 1 regarding the beta parameter, but it is not used. Should it be table 2?

---

> ### Author Response · Authors · 2020-11-25
> **We would like to thank AnonReviewer4 for comments on our paper and highlighting the replicability of our method for other researchers.**
>
> We would like to thank AnonReviewer4 for comments on our paper and highlighting the replicability of our method for other researchers.
>
> “Experiments ”: The benefit of the method is that, under uncertain (real-world) conditions, we are able to produce an order of magnitude reduction in average confidence when no pertinent objects are present in a given image. On a given dataset, context helps traditional classifiers make correct predictions, but for real-world applications, such reliance can result in harmful effects (as in the case of autonomous driving). Please see section A.6, our approach produces extremely low confidence values when presented with background images.
>
> “underconfidence”: Underconfidence measures how close the confidence of a correct prediction is to a value of 1. As our confidence is grounded in the size of the object being predicted, our underconfidence is higher and our values are close to the objectness score. Underconfidence has no effect on accuracy, as for a classifier the maximum value of the output is used to assign the class label.
>
> “object detection”: We use different pre-trained classifiers as the backbone for FRCNN. We do not apply ALS directly to the object detections task, but evaluate the our ImageNet learnt features to other baselines. Our AP is 0.001 lower compared to CutMix (state of the art using ResNet50). However CutMix is also able to produce an accuracy of 9.2% using pure context. We perform much better under uncertain conditions, while remaining almost as good as CutMix for object detection. We don’t measure confidence for the transfer learning portion. We replace the backbone and fine-tune to show the effectiveness of the representation that we learn using our approach of Adaptive Label Smoothing.
>
> “The method needs the amount of percentage of pixels of the object to adjust the labels.”: While objectness can be computed using other approaches (as pointed out by R2 and R3), we use bounding boxes as they are available for the ImageNet dataset to implement our adaptive label smoothing approach. We are not relying on object coordinates, but we use the information to compute foreground vs. background ratio. Our beta parameter can be used to control the amount of context dependence. We gain 2.1mAP compared to hard labels using the learnt representation.
>
> “OpenImages”: We originally left out the discussion of OpenImages due to lack of space, but have now added those results (classification and transfer learning results) to the appendix. We describe MS COCO more completely due to its popularity and standard splits.
>
> “experimental setting”: We are interested in the strict definition of a classifier, and to that end we limit the number of objects to 1 per image as is common (with MNIST, CIFAR, etc.). If we have bounding box annotation for multiple objects, we mask all but one object with pixel means. As some ImageNet images have multiple objects, we end up with 54K repeated images with different objects in them. Please see section A.2 for more information.
>
> “entropy: A very confident prediction (peaky distribution) will have low entropy and a less confident prediction (uniform distribution) will have a high entropy. Computing entropies of the predictions and their use for fine-grained out of distribution is being explored by the authors.
>
> “Discussion in 4.2 does not discuss the other metrics ECE, U.conf, MCE enough to give some insights about the results. In particular the U.conf gets higher with the method. What is happening and how does this affects the confidence?”: We have updated the discussion in the paper. Please see section 4.2 in the paper. Underconfidence has no effect on accuracy as for a classifier the maximum value of the output is used to assign the class label.
>
> “It is not clear in table 2 what are the two experiments with 0.474M pre train N and what information they give.”: We have streamlined the section and moved the results on different splits to the appendix. We rely on objectness scores for our approach. About 0.474M images from the ImageNet training set have bounding box annotations, and some of these images have multiple objects. By masking all but one object we end up with 0.528M images of different objects. In the main paper we are discussing classifiers trained using only the 0.528M split. Please refer to table 5 in the appendix for results using different versions of the ImageNet dataset. Also refer to section A.2 for more information.
>
> “Sec 4.4 mentions table 1 regarding the beta parameter, but it is not used. Should it be table 2?”: Thank you for catching that mistake in our original submission, in which that table was omitted due to lack of space. Those results are discussed in the updated version of the paper.
>
> We have updated the submission and added the requested information. We hope AnonReviewer4 will take these changes into account. We welcome additional suggestions for our camera-ready version as well.

---

### Decision · Program_Chairs · 2021-01-07
**Final Decision**

**Decision:**

Reject

**Comment:**

The paper introduces an adaptive label smoothing technique, where the smoothing factor is computed based on the relative object size within an image, in order to address the problem of overconfident predictions. All reviewers recommend rejection based on limited technical contribution and unclear benefits of the proposed method. During the rebuttal phase, the authors carried out more experiments and clarified several other questions asked by the reviewers. The response was well received, but did not eliminate the main concerns about the paper. While the idea is interesting and has potential, the AC agrees with the reviewers that the paper is not ready for ICLR, and encourages the authors to improve the paper according to the reviews and submit it to another top conference.